# Diagnostic accuracy of a smartphone-based device (VistaView) for detection of diabetic retinopathy: A prospective study

**Rida Shahzad[1], Arshad Mehmood[1], Danish Shabbir[1], M. A. Rehman Siddiqui** [1,2]*

**1** Shahzad Eye Hospital, Karachi, Pakistan, **2** Department of Ophthalmology and Visual Sciences, Aga Khan University, Karachi, Pakistan

* rehman.siddiqui@gmail.com

**Data Availability Statement:** All relevant data are within the manuscript and its Supporting Information files.

## Abstract

### Background

Diabetic retinopathy (DR) is a leading cause of blindness globally. The gold standard for DR screening is stereoscopic colour fundus photography with tabletop cameras. VistaView is a novel smartphone-based retinal camera which offers mydriatic retinal imaging. This study compares the diagnostic accuracy of the smartphone-based VistaView camera compared to a traditional desk mounted fundus camera (Triton Topcon). We also compare the agreement between graders for DR screening between VistaView images and Topcon images.

### Methodology

This prospective study took place between December 2021 and June 2022 in Pakistan. Consecutive diabetic patients were imaged following mydriasis using both VistaView and Topcon cameras at the same sitting. All images were graded independently by two graders based on the International Classification of Diabetic Retinopathy (ICDR) criteria. Individual grades were assigned for severity of DR and maculopathy in each image. Diagnostic accuracy was calculated using the Topcon camera as the gold standard. Agreement between graders for each device was calculated as intraclass correlation coefficient (ICC) (95% CI) and Cohen's weighted kappa (k).

### Principal findings

A total of 1428 images were available from 371 patients with both cameras. After excluding ungradable images, a total of 1231 images were graded. The sensitivity of VistaView for any DR was 69.9% (95% CI 62.2–76.6%) while the specificity was 92.9% (95% CI 89.9–95.1%), and PPV and NPV were 80.5% (95% CI 73–86.4%) and 88.1% (95% CI 84.5–90.9) respectively. The sensitivity of VistaView for RDR was 69.7% (95% CI 61.7–76.8%) while the specificity was 94.2% (95% CI 91.3–96.1%), and PPV and NPV were 81.5% (95% CI 73.6–87.6%) and 89.4% (95% CI 86–92%) respectively. The sensitivity for detecting maculopathy in VistaView was 71.2% (95% CI 62.8–78.4%), while the specificity was 86.4% (82.6–89.4%). The PPV and NPV of detecting maculopathy were 63% (95% CI 54.9–70.5%) and

**Funding:** The author(s) received no specific funding for this work.

**Competing interests:** The authors have declared that no competing interests exist.

90.1% (95% CI 86.8–92.9%) respectively. For VistaView, the ICC of DR grades was 78% (95% CI, 75–82%) between the two graders and that of maculopathy grades was 66% (95% CI, 59–71%). The Cohen's kappa for retinopathy grades of VistaView images was 0.61 (95% CI, 0.55–0.67, p<0.001), while that for maculopathy grades was 0.49 (95% CI 0.42–0.57, p<0.001). For images from the Topcon desktop camera, the ICC of DR grades was 85% (95% CI, 83–87%), while that of maculopathy grades was 79% (95% CI, 75–82%). The Cohen's kappa for retinopathy grades of Topcon images was 0.68 (95% CI, 0.63–0.74, p<0.001), while that for maculopathy grades was 0.65 (95% CI, 0.58–0.72, p<0.001).

## Conclusion

The VistaView offers moderate diagnostic accuracy for DR screening and may be used as a screening tool in LMIC.

### Author summary

Diabetic retinopathy (DR) is a highly prevalent retinal disease globally which can lead to irreversible loss of vision if left untreated. Therefore, it is essential that efficient systematic screening processes be established to facilitate timely diagnosis and management to prevent loss of vision. Standard methods of DR screening require heavy and expensive equipment operated by trained professionals and are often inaccessible to marginalised communities. In this study, we investigated the diagnostic accuracy of a lightweight, portable and relatively inexpensive smartphone-based retinal camera to detect DR, compared to a standard tabletop imaging device. We found that diagnostic accuracy needs to improve further to make these devices a suitable option for DR screening, especially in low- and middle-income countries where access to healthcare has several barriers.

## Introduction

Diabetic retinopathy (DR) is a leading cause of blindness globally within the working age group [1]. The prevalence of diabetes mellites is increasing exponentially every year especially in low-middle income countries (LMIC) including Pakistan and India. Consequently the burden of DR is also rising, and its global prevalence is estimated to reach 160.5 million by 2045 [2]. DR remains asymptomatic until it reaches an advanced stage when loss of vision occurs. To prevent blindness, it is essential that DR is detected and treated in its early stages, as timely management can reduce the risk of severe visual loss by up to 90% [3]. Detection of DR is conventionally done through retinal screening in diabetic patients by ophthalmologists or colour fundus photography [4,5]. The gold standard for DR screening is seven field stereoscopic colour fundus photography. This requires trained personnel and bulky, immovable desk-mounted fundus cameras, which are costly and not universally accessible in the community. However, single-field posterior fundus imaging is reported to be as accurate for the screening of DR [6,7].

Recently smartphone-based retinal photography to screen DR has gained traction. This modality has the potential to be efficient in terms of time, cost, and space [8–10]. VistaView is a smartphone-based, portable, handheld camera manufactured by Volk Optical Inc (Mentor,

Ohio) which enables quick mydriatic retinal imaging. It captures single-field 55 degrees fundus images with a resolution of 28.4 pixels/degree which can be instantly viewed and analysed.

The aim of this study was to evaluate the diagnostic accuracy of VistaView (the index test) with a standard Topcon Triton (Topcon, Tokyo, Japan) desk mounted fundus camera for the detection of diabetic retinopathy.

## Materials and Methods

This prospective, cohort study was carried out at a tertiary eye hospital in Karachi, Pakistan between December 2021 to June 2022. We recruited 375 consecutive patients (714 eyes) attending the eye clinic with known type-1 or type-2 diabetes above the age of 16 years. Exclusion criteria was pre-diabetics, and patients who were previously treated for PDR with laser or vitrectomy, as these patients would have retinal scars from treatment and likely to be aware of the advanced stage of DR and/ or already under a hospital eyecare service.

This manuscript followed the Standards for Reporting of Diagnostic Accuracy Studies (STARD) 2015 guidelines [11]. Ethics review board of the hospital approved the study (Ref# 605685). Informed written consent was obtained from all patients. The research adhered to the tenets of Helsinki. We collected demographic information from each patient including age, gender, and duration of the disease. Each recruited patient underwent dilation with Tropicamide 1% and Cyclopentoate Hydrochloride 1%, and colour fundus photography of both eyes was performed by trained technicians using the standard Topcon camera and the handheld VistaView device at the same sitting in a random sequence. Topcon fundus camera captures high resolution fundus photographs enhanced by PixelSmart technology. It offers up to 45 degrees field of view. Its resolution on fundus is 60 Lines/mm in the center, 40 lines/mm in the middle, and 25 Lines/mm in the periphery. The VistaView acquires images with an output resolution of 3072 x 2122 pixels with a field of view of up to 55 degrees. All images were anonymized, and uploaded to a protected cloud database.

### Diabetic retinopathy grading

Images acquired from both devices were analysed and graded independently by two certified graders using the ICDR (International Classification of Diabetic Retinopathy) system [12]. The DR stages included: no retinopathy, mild nonproliferative DR(NPDR), moderate NPDR, severe NPDR, and PDR (Fig 1). The images were also graded for presence or absence of maculopathy (exudates, hemorrhages or apparent thickening within 1 disc diameter from the fovea). The images acquired using VistaView were graded before the Topcon images to avoid information bias. Consensus was reached for any disagreements. If required, arbitration was done by a fellowship trained vitreo-retinal specialist for final grades. The final DR grades assigned to images acquired with the desktop Topcon camera were considered as the gold standard.

### Image quality

The photographs taken with both the cameras were graded on a 5-level grading scheme as follows: [13,14] (Fig 2)

- Grade 0: Ungradable (no retinal details visible due to media opacities such as dense cataract)

- Grade 1: Poor (only gross retinal changes visible such as hemorrhages and dense hard exudates)

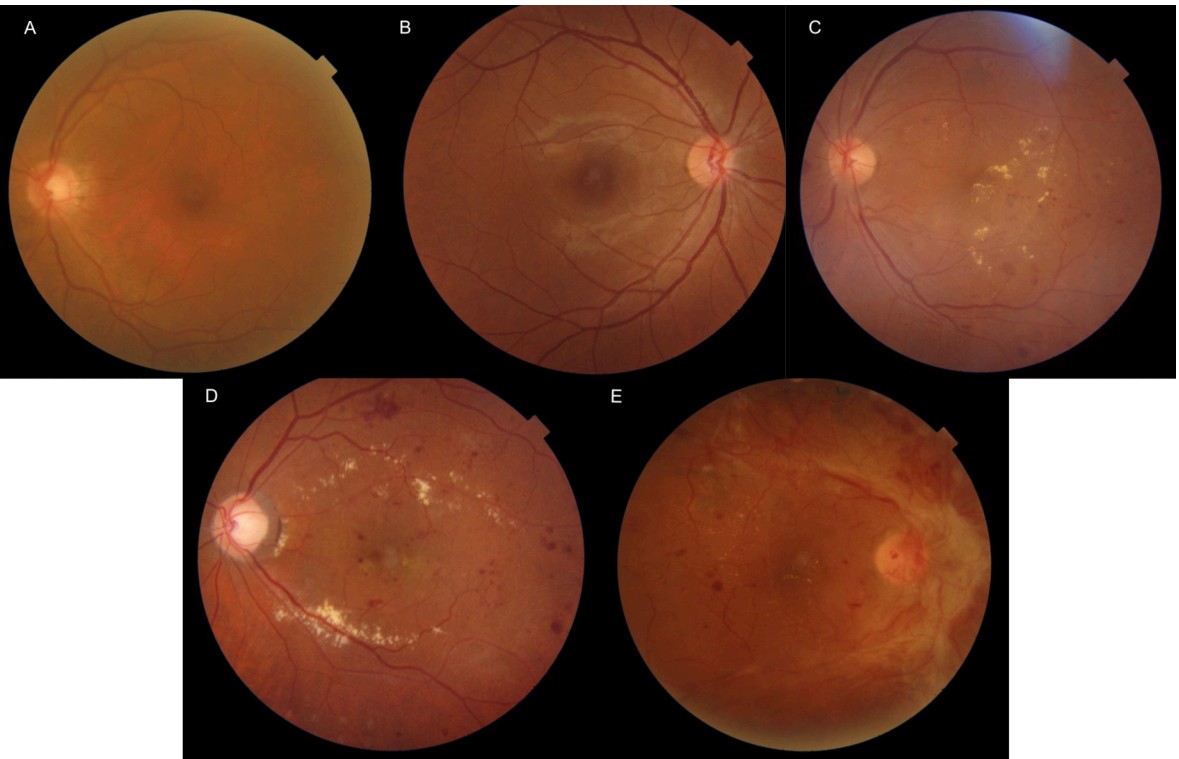

**Fig 1. Examples of stages of diabetic retinopathy based on ICDR system.** (A) No DR; (B) Mild NPDR; (C) Moderate NPDR; (D) Severe NPDR; (E) PDR. DR, diabetic retinopathy; NPDR, nonproliferative diabetic retinopathy; PDR, proliferative diabetic retinopathy. ICDR, International Classification of Diabetic Retinopathy.

- Grade 2: Satisfactory (major retinopathy details visible; minor degrees of retinopathy and subtle new vessels not clearly detectable.

- Grade 3: Good (most of retinopathy changes clear)

- Grade 4: Excellent (all lesions clearly visible)

## Statistical analysis

Continuous variables such as age and gender of patients were reported as means (+/- standard deviation). Statistical analysis was performed using SPSS (IBM, SPSS Inc.) version 27. The sensitivity and specificity of the VistaView was calculated, against Topcon Triton fundus image (the gold standard). Diagnostic accuracy was evaluated at 2 dichotomous values for any DR and referable DR (RDR). RDR was defined as moderate NPDR stage or higher (ICDR>1) and/or maculopathy. The level of agreement between the two graders was assessed using the Cohen's kappa statistic for both Topcon images and VistaView images. Inter-rater reliabilities between the two graders were calculated as ICCs (intra-class coefficients). Gradeability of images was reported as descriptive frequencies.

## Results

### Demographics

A total of 371 patients were enrolled in the study. Of these, 194 (52%) were male. The mean age of the overall cohort was 59 (Table 1). The mean duration of diabetes in the study population was 10.5 years.

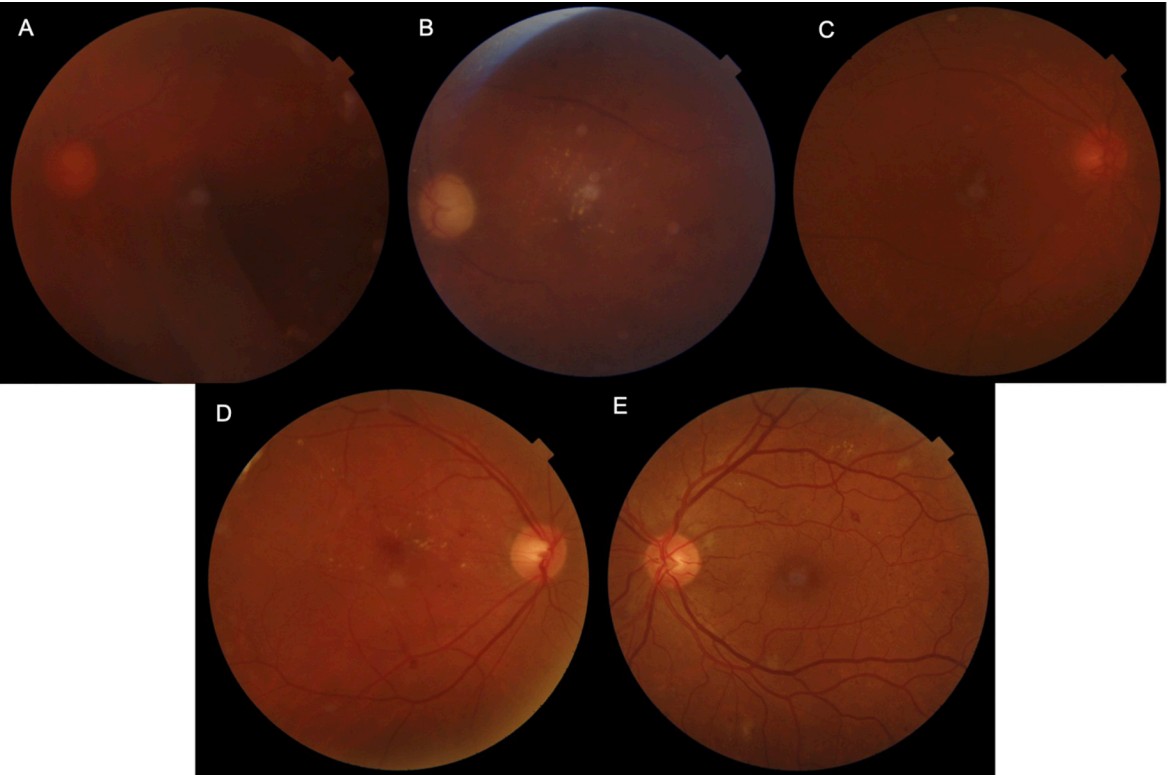

**Fig 2. Examples of image quality grades.** (A) Grade 0; (B) Grade 1; (C) Grade 2; (D) Grade 3; (D) Grade 4.

## Gradeability

A total of 1428 images were available from 371 patients from both the cameras (Fig 3). A total of 1231 images were graded. Numbers and percentages of retinopathy and maculopathy grades detected with each device are given in Table 2.

## Diagnostic accuracy

The overall sensitivity of VistaView for any DR was 69.9% (95% CI 62.2–76.6%) while the specificity was 92.9% (95% CI 89.9–95.1%), (Table 3). PPV and NPV were 80.5% (95% CI 73–86.4%) and 88.1% (95% CI 84.5–90.9) respectively. The receiver operating curve (ROC) for any DR is given in Fig 4. The sensitivity of VistaView for RDR was 69.7% (95% CI 61.7–76.8%) while the specificity was 94.2% (95% CI 91.3–96.1%). PPV and NPV were 81.5% (95% CI 73.6–87.6%) and 89.4% (95% CI 86–92%) respectively. The sensitivity for detecting

**Table 1. Baseline characteristics.**

| Categorical variables | Frequency (%) | |
|---|---|---|
| Sex | | |
| Male | 194 (52.2) | |
| Female | 177 (47.7) | |
| **Continuous variables** | **Mean** | **Range** |
| Age (years) | 59 | 22–84 |
| Duration of diabetes (years) | 10.5 | 1–40 |

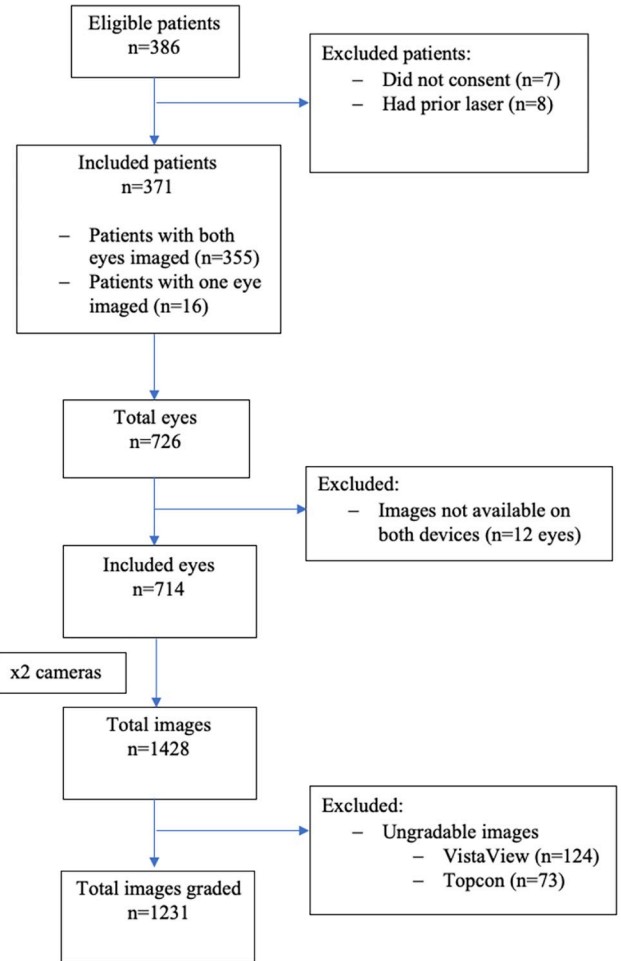

**Fig 3. Flow chart of data collection.**

maculopathy in VistaView was 71.2% (95% CI 62.8–78.4%), while the specificity was 86.4% (82.6–89.4%). The PPV and NPV of detecting maculopathy were 63% (95% CI 54.9–70.5%) and 90.1% (95% CI 86.8–92.9%) respectively (Table 3).

**Table 2. Retinopathy and maculopathy stages detected with both devices with percentages.** NPDR: nonproliferative diabetic retinopathy, PDR: proliferative diabetic retinopathy, M: maculopathy, U: ungradable.

|  | VistaView n (%) | Topcon n (%) |
|---|---|---|
| No retinopathy (S0) | 429 (60%) | 462 (64.7%) |
| Mild NPDR (S1) | 14 (2%) | 15 (2.1%) |
| Moderate NPDR (S2) | 112 (15.7%) | 135 (19%) |
| Severe NPDR (S3) | 8 (1.1%) | 7 (1%) |
| PDR (S4) | 18 (2.5%) | 21 (3%) |
| U | 133 (18.6%) | 73 (10.2%) |
| **M grade** |  |  |
| Maculopathy present (M1) | 418 (58.5%) | 491 (68.8%) |
| Maculopathy absent (M0) | 166 (23.2%) | 150 (21%) |
| U | 130 (18.2%) | 73 (10.2%) |

**Table 3. Diagnostic accuracy for retinopathy and maculopathy detection by VistaView in comparison to Topcon.** DR: diabetic retinopathy, RDR: referrable diabetic retinopathy.

| | Sensitivity % (95% CI) | Specificity % (95% CI) | PPV (95% CI) | NPV (95% CI) | False positive rate % | False negative rate % |
|---|---|---|---|---|---|---|
| **DR** | | | | | | |
| Any DR | 69.9 (62.2–76.6) | 92.9 (89.9–95.1) | 80.5 (73–86.4) | 88.1 (84.5–90.9) | 7.1 | 30.1 |
| RDR | 69.7 (61.7–76.8) | 94.2 (91.3–96.1) | 81.5 (73.6–87.6) | 89.4 (86–92) | 5.8 | 30.3 |
| **Maculopathy** | 71.2 (62.8–78.4) | 86.4 (82.6–89.4) | 63 (59.4–70.5) | 90.1 (86.8–92.9) | 13.6 | 28.8 |

## Agreement

For VistaView, the ICC of DR grades was 78% (95% CI, 75–82%) between the two graders and that of maculopathy grades was 66% (95% CI, 59–71%). The Cohen's kappa for retinopathy grades of VistaView images was 0.61 (95% CI, 0.55–0.67, p<0.001), while that for maculopathy grades was 0.49 (95% CI 0.42–0.57, p<0.001). For images from the Topcon desktop camera, the ICC of DR grades was 85% (95% CI, 83–87%), while that of maculopathy grades was 79% (95% CI, 75–82%). The Cohen's kappa for retinopathy grades of Topcon images was 0.68 (95% CI, 0.63–0.74, p<0.001), while that for maculopathy grades was 0.65 (95% CI, 0.58–0.72, p<0.001). (Fig 5, Table 4)

## Image quality

Image quality scores are summarized in Table 5. For VistaView images, grader 1 scored 83 images (11.6%) as ungradable, 85 images (12%) as poor, 247 (34.6%) as satisfactory, 233 (32.6%) as good, and 66 (9.2%) as excellent. Grader 2 scored 106 images (14.8%) as ungradable, 261 images (36.5%) as poor, 258 images (35.9%) as satisfactory, 77 images (10.8%) as good, and 12 (1.7%) as excellent.

For the desktop camera images, grader 1 scored 56 images (7.8%) as ungradable, 66 images (9.2%) as poor, 123 images (17.2%) as satisfactory, 273 images (38.2%) as good, and 196 (27.4%) as excellent. Grader 2 scored 51 images (7.1%) as ungradable, 35 images (4.9%) as poor, 37 images (5.1%) as satisfactory, 155 images (21.7%) as good, and 436 (61.1%) as excellent.

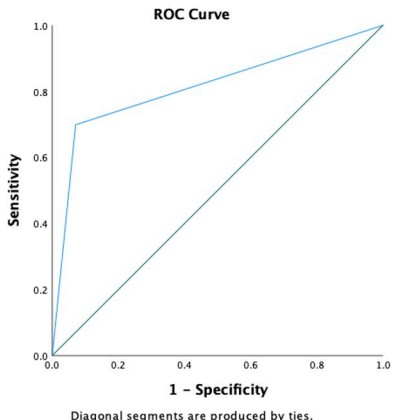

**Fig 4. ROC curve of any DR and RDR.**

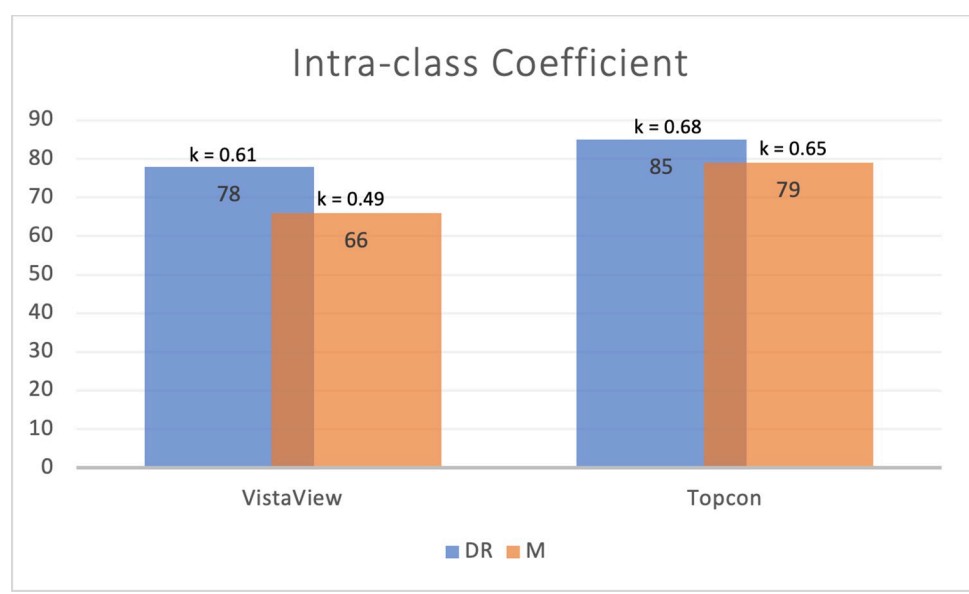

**Fig 5. Agreement metrics of graders for both devices.** DR, diabetic retinopathy; M, maculopathy.

## Discussion

This study evaluates the diagnostic accuracy of VistaView, a smartphone-based fundus camera for the detection of diabetic retinopathy. To the best of our knowledge, there are no diagnostic accuracy studies of VistaView which evaluate it against a gold standard for detecting DR. We obtained sensitivity of 69.9% and specificity of 92.9% for any DR. For RDR, the sensitivity and specificity were 69.7% and 94.2% respectively. Our findings corroborate with previously published reports in the literature validating the use of smartphone-based devices for screening of DR, as shown in a recent meta-analysis done by Tan et al. [15] They reported pooled sensitivities and specificities of 87% and 94% for smartphone-based devices for any DR; 91% and 89% for RDR and 79% and 93% for diabetic macular edema respectively; when the gold standard was colour fundus photos. Studies comparing such devices with clinical examination have also found high sensitivities and specificities. Sengupta et al. described the sensitivity and specificity of smartphone-based retinal imaging for DR screening compared to dilated fundus examination to be approximately 93% and 89% respectively [16]. Zhang also reported comparable findings in a similar study done earlier [17]. The British Diabetic Association has established the cut-offs of 80% sensitivity and 95% specificity for a viable screening system. National Institute for Clinical Excellence (NICE) guidelines recommend similar standards for sensitivity and specificity [18]. Although our study showed sensitivity lower than these cut-offs due to some limiting factors of the VistaView discussed hereafter, the specificity we obtained met these criteria–a high specificity reduces the burden on hospital eye services for patients who need to seek specialist care for positive tests.

**Table 4. Interrater reliability expressed as ICC (intra-class coefficient) and Cohen's kappa agreement.**

| | VistaView % (95% CI) | | Topcon % (95% CI) | |
|---|---|---|---|---|
| | DR grades | M grades | DR grades | M grades |
| ICC | 78 (75–82) | 66 (59–71) | 85 (83–87) | 79 (75–82) |
| k agreement | 0.61 (0.55–0.67) (p<0.001) | 0.49 (0.42–0.57) | 0.68 (0.63–0.74) (p<0.001) | 0.65 (0.58–0.72) |

**Table 5. Quality of images with both fundus cameras.**

| Image quality | Grader 1 | | | | Grader 2 | | | |
| --- | --- | --- | --- | --- | --- | --- | --- | --- |
| | VistaView | | Topcon | | VistaView | | Topcon | |
| | n | % | n | % | n | % | n | % |
| Excellent | 66 | 9.2 | 196 | 27.4 | 12 | 1.7 | 436 | 61.1 |
| Good | 233 | 32.6 | 273 | 38.8 | 77 | 10.8 | 155 | 21.7 |
| Satisfactory | 247 | 34.6 | 123 | 17.2 | 258 | 35.9 | 37 | 5.1 |
| Poor | 85 | 12 | 66 | 9.2 | 261 | 36.5 | 35 | 4.9 |
| Ungradable | 83 | 11.6 | 56 | 7.8 | 106 | 14.8 | 51 | 7.1 |

Image quality can be a limiting factor to the usefulness of smartphone-based DR screening systems. Poor image quality may lead to ungradable images. Eighteen percent of the VistaView images were labelled as ungradable. Gradeability also depends on other factors such as presence of media opacities which can lower the overall diagnostic accuracy of the device. The percentage of ungradable desktop camera images (10.2%) was markedly less than that of smartphone based camera (18.6%). The reason for poor quality images includes low internal resolution of the device, and the focusing process of handheld cameras. These factors may contribute to the VistaView's sensitivity being lower than that of other handheld cameras in the literature, and lower than the recommended 80%. Additionally, it may be difficult to achieve well focused images with smartphone-based devices because, unlike standard desktop cameras, smartphone devices do not come with a built-in chin stabilizer. It is worth noting that some studies in the literature which have reported higher diagnostic accuracy metrices compared to the VistaView have excluded patients with media opacities [8,13,19]. Our study includes such patients which makes it more representative of real-world settings. Interestingly, both certified graders had good agreement scores (k>0.60) for retinopathy grades with VistaView images, (k = 0.61) which were comparable to agreement levels using the standard desktop camera. (k = 0.68)

Patient preference and comfort is an important factor to consider when developing a pathway for DR screening. Studies have reported acceptable patient comfort with smartphone-based fundus cameras due to lower light intensity of the LED, compared to the high intensity flash in the traditional fundus camera [19]. Design simplicity and portability are additional factors which contribute to higher general acceptability among patients. These factors may improve compliance with screening and improve overall effectiveness of such a system. In future studies, patient satisfaction with VistaView smartphone-based camera may be explicitly evaluated.

Our technical staff received comprehensive training on the VistaView before initiating retinal imaging for data collection. Studies have shown that duration of training with smartphone-based fundus cameras for screening DR has an association with higher image quality and reduced examination time [20–22].

Vision loss has significant financial and economic implications especially in LMICs such as Pakistan. Because LMICs contribute to majority of the world's diabetic population [23], systematic screening needs be implemented to prevent vision loss secondary to DR in these countries. Smartphone-based screening of DR is particularly valuable in these regions, where there are several barriers to availability and access to health services, including cost, poor infrastructure, lack of equipment, and deficiencies in trained personnel. Smartphone-based fundus imaging has the potential to lower the burden of DR screening on hospital eye services, particularly in LMICs.

Studies show that eye care is imperative to achieve The United Nations 2030 Sustainable Development Goals (SDGs). Improving access to eyecare will help achieve many of these

SDGs including reduction of poverty, and increased economic productivity, educational performance, and equity [24]. The World Health Organization (WHO) and International Diabetes Federation have acknowledged the role of low-cost smartphone-based devices for DR screening especially by non-physicians [23,25–27]. This can be attributed to their cost-effectiveness, portability, low computational power and space requirements, ease of use, and less training requirement [28]. Tele-ophthalmology may benefit from these attributes of the Vista-View to provide efficient and cost-effective DR screening programmes. When introduced at the primary care level, effective screening may be delivered at the point-of-care and the risk of vision loss secondary to DR may be significantly lowered.

Recent advances in technology have allowed integration of artificial intelligence (AI) into DR screening systems. AI algorithms trained to make automated diagnoses of DR on retinal images acquired from smartphone-based fundus cameras have shown to have high diagnostic accuracy in high prevalence settings when compared to human grading [29–31]. This can potentially replace the need for human graders' classification of DR in screening programmes. In future, this could diminish the burden of DR screening on ophthalmologists, and their expertise can be redirected to appropriately managing advance disease.

There were certain limitations in our study. Firstly, requirement of mydriasis adds to the overall acquisition time for imaging and is associated with the side effect of blurry vision for several hours. Another limitation was the use of convenience sampling as all recruited participants were attending a tertiary care eye hospital. Therefore, results may not be generalizable to community settings. Lastly, we did not perform patient acceptability survey in our study.

In conclusion, DR screening with smartphone-based fundus cameras is a potential option for systematic screening of DR. Further studies are needed to evaluate sensitivity and specificity in various populations.

## Supporting information

**S1 File. Infographic: Diagnostic accuracy of VistaView for diabetic retinopathy detection.**
(PDF)

**S2 File. STARD checklist: Completed STARD checklist.**
(PDF)

## Acknowledgments

We would like to acknowledge the contributions of the doctors of Shahzad Eye Hospital–Dr. M. H. Shahzad and Dr. Harris Shahzad for referring diabetic patients for DR screening.

## Author Contributions

**Conceptualization:** Rida Shahzad, M. A. Rehman Siddiqui.

**Data curation:** Rida Shahzad.

**Formal analysis:** Rida Shahzad, M. A. Rehman Siddiqui.

**Investigation:** Rida Shahzad, Arshad Mehmood, Danish Shabbir, M. A. Rehman Siddiqui.

**Methodology:** Rida Shahzad, M. A. Rehman Siddiqui.

**Project administration:** Rida Shahzad.

**Resources:** M. A. Rehman Siddiqui.

**Software:** Danish Shabbir.

**Supervision:** Rida Shahzad, M. A. Rehman Siddiqui.

**Validation:** Rida Shahzad.

**Writing – original draft:** Rida Shahzad.

**Writing – review & editing:** M. A. Rehman Siddiqui.

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
