## [Decision Letter · Decision Letter 0]

9 Jul 2024

PDIG-D-24-00116

Diagnostic Accuracy of a Smartphone-Based Device (VistaView) for Detection of Diabetic Retinopathy: A Prospective Study

PLOS Digital Health

Dear Dr. Siddiqui,

Thank you for submitting your manuscript to PLOS Digital Health. After careful consideration, we feel that it has merit but does not fully meet PLOS Digital Health's publication criteria as it currently stands. Therefore, we invite you to submit a revised version of the manuscript that addresses the points raised during the review process.

Please submit your revised manuscript within 60 days Sep 07 2024 11:59PM. If you will need more time than this to complete your revisions, please reply to this message or contact the journal office at digitalhealth@plos.org. Please include the following items when submitting your revised manuscript:

We look forward to receiving your revised manuscript.

Kind regards,

Luis Filipe Nakayama, M.D.

Academic Editor

PLOS Digital Health

Journal Requirements:

1. We ask that a manuscript source file is provided at Revision. Please upload your manuscript file as a .doc, .docx, .rtf or .tex.

2. Please provide separate figure files in .tif or .eps format.

Additional Editor Comments (if provided):

Review of “Diagnostic Accuracy of a Smartphone-Based Device (VistaView) for Detection of Diabetic Retinopathy: A Prospective Study”

* I suggest a review of English and text flow. 

* Line 162: Is 59 the mean age of the males specifically or the overall cohort?

* Comorbidities: Are there any reported comorbidities among the participants?

* Capturing Protocol: What was the image-capturing protocol? Was it 1 image or 2 images per eye?

 * How was the issue of a single bad-quality image in a set handled?

 * How was the labeling performed—at the image level, eye level, or patient level?

* Referable Criteria: Please clarify the referable criteria. Is it ICDR >1 and/or macular edema?

* Handling Low-Quality Images: How were low-quality images managed between devices for statistical analysis?

 * It would be beneficial to describe the quality parameters across demographics, diabetic retinopathy (DR), and maculopathy severity between devices.

* I suggest reporting the results for the ICDR scores grading between the devices, as the labeling grading was conducted instead of just any DR and Referable DR.

* Image Quality Reporting: Why were image quality reports done by individual graders instead of by consensus and adjudication, as with DR scores?

* False Positives and False Negatives: Please report the false positive and false negative rates for each classification.

* Additionally, I recommend including a statistical analysis to compare the performance of the devices, demographic, and associated factors.

* How do you justify the low low sensitivity of this device? I suggest expanding the discussion.

Reviewers' comments:

Reviewer's Responses to Questions

**Comments to the Author**

1. Does this manuscript meet PLOS Digital Health’s publication criteria? Is the manuscript technically sound, and do the data support the conclusions? The manuscript must describe methodologically and ethically rigorous research with conclusions that are appropriately drawn based on the data presented.

Reviewer #1: Partly

Reviewer #2: Partly

2. Has the statistical analysis been performed appropriately and rigorously?

Reviewer #1: Yes

Reviewer #2: No

3. Have the authors made all data underlying the findings in their manuscript fully available (please refer to the Data Availability Statement at the start of the manuscript PDF file)?

Reviewer #1: Yes

Reviewer #2: Yes

4. Is the manuscript presented in an intelligible fashion and written in standard English?

Reviewer #1: No

Reviewer #2: Yes

5. Review Comments to the Author

Reviewer #1: Shahzad and colleagues present an interesting and well-written manuscript entitled "Diagnostic Accuracy of a Smartphone-Based Device (VistaView) for Detection of Diabetic Retinopathy: A Prospective Study". The authors conducted a prospective cohort study comparing the diagnostic accuracy of a handheld smartphone based fundus camera in screening for diabetic retinopathy against the gold standard, a standard table-top camera. Sensitivities of around 70% and specificities of about 90% were found. The inter-observer agreement was markedly lower for the handheld camera, as well as the proportion of ungradeable images.

Overall this is a rigourosly conducted study and the topic is timely and relevant. The findings are important for the scientific community.

A few issues should be addressed before the paper can be evaluated further:

1) Although the sensitivities found by the authors are lower than the recommended 80%, the authors conclude that "The vistaview offers high diagnostic accuracy for DR screening and can be used as a screening tool in LMIC". This conclusion is not based on the data. Please reconsider.

2) The sensitivities reported here are lower than for other handheld cameras in the literature. Please discuss.

3) The discussion is a bit lengthy and could be shortened.

4) Suggest to present the results on image gradeability before the accuracy data.

5) The manuscript would benefit from native speaker English review.

Reviewer #2: The objective of this article is to evaluate the diagnostic accuracy of VistaView in the screening of diabetic retinopathy (DR). This clarity helps readers quickly grasp the core content and research focus of the article. The article employs scientific research methods, including a prospective study design, reasonable sample selection, and accurate evaluation indicators. The selection of these methods ensures the reliability and effectiveness of the study. After a careful reading and review of the full text of the paper, the reviewer proposes the following comments:

1. Add a Detailed Description of the Research Object: It is recommended to add a detailed description of the research object in the research methods section, including key information such as inclusion and exclusion criteria, so that readers can have a more comprehensive understanding of the characteristics and representativeness of the research object.

2. Discuss the Limitations and Future Directions of the Research: In the discussion section, it is recommended that the author discuss the limitations of this study and possible future research directions. For example, the applicability of VistaView in different populations, comparative studies with other screening tools, and DR screening methods based on artificial intelligence technology can be explored. This helps readers to have a more comprehensive understanding of the value and significance of this study and provides useful references for future research.

3. All Subjects Underwent Mydriasis for Standard Color Fundus Photography: What is the significance of mydriasis? What is the value of promoting this method?

4. Clarification of "Maculopathy": Does “Maculopathy” refer to diabetic macular edema (DME) or does it include all macular lesions?

5. Terminology: The term “colour fundus photography” should not be abbreviated to “fundus photography.”

6. PLOS authors have the option to publish the peer review history of their article (what does this mean?). If published, this will include your full peer review and any attached files.

**Do you want your identity to be public for this peer review?** For information about this choice, including consent withdrawal, please see our Privacy Policy.

Reviewer #1: No

Reviewer #2: Yes: Weihua Yang

---

## [Decision Letter · Decision Letter 1]

20 Sep 2024

Diagnostic Accuracy of a Smartphone-Based Device (VistaView) for Detection of Diabetic Retinopathy: A Prospective Study

PDIG-D-24-00116R1

Dear Dr. Siddiqui,

We are pleased to inform you that your manuscript 'Diagnostic Accuracy of a Smartphone-Based Device (VistaView) for Detection of Diabetic Retinopathy: A Prospective Study' has been provisionally accepted for publication in PLOS Digital Health.

Best regards,

Luis Filipe Nakayama, M.D.

Academic Editor

PLOS Digital Health

All my comments have been successfully addressed.

Reviewer Comments (if any, and for reference):

Reviewer's Responses to Questions

**Comments to the Author**

1. If the authors have adequately addressed your comments raised in a previous round of review and you feel that this manuscript is now acceptable for publication, you may indicate that here to bypass the “Comments to the Author” section, enter your conflict of interest statement in the “Confidential to Editor” section, and submit your "Accept" recommendation.

Reviewer #1: All comments have been addressed

Reviewer #2: All comments have been addressed

2. Does this manuscript meet PLOS Digital Health’s publication criteria? Is the manuscript technically sound, and do the data support the conclusions? The manuscript must describe methodologically and ethically rigorous research with conclusions that are appropriately drawn based on the data presented.

Reviewer #1: Yes

Reviewer #2: Yes

3. Has the statistical analysis been performed appropriately and rigorously?

Reviewer #1: Yes

Reviewer #2: Yes

4. Have the authors made all data underlying the findings in their manuscript fully available (please refer to the Data Availability Statement at the start of the manuscript PDF file)?

Reviewer #1: No

Reviewer #2: Yes

5. Is the manuscript presented in an intelligible fashion and written in standard English?

Reviewer #1: Yes

Reviewer #2: Yes

6. Review Comments to the Author

Reviewer #1: (No Response)

Reviewer #2: I am satisfied with the authors’ corrections.

7. PLOS authors have the option to publish the peer review history of their article (what does this mean?). If published, this will include your full peer review and any attached files.

**Do you want your identity to be public for this peer review?** For information about this choice, including consent withdrawal, please see our Privacy Policy.

Reviewer #1: No

Reviewer #2: **Yes: **Weihua Yang
